# OpenReview forum: "LlavaGuard: An Open VLM-based Framework for Safeguarding Vision Datasets and Models"
_ICML.cc/2025/Conference — ICML 2025 poster_

### Official Review · Reviewer_LJnY · 2025-02-19

**Overall Recommendation:** 4

**Summary:**

This paper introduces LlavaGuard, a suite of vision safeguards. They decribe a systematic framework including safety taxonomy, data preprocessing, augmentation, and training setup. Then they build a multimodal safety dataset and train LlavaGuard models on this. Through extensive experiments, they demonstrate that LlavaGuard outperforms previous SOTA methods and is applicable to several real world problems.

## update after rebuttal

I'm satisfied with the rebuttal. With the additional experiments, the paper is further strengthened.

**Claims And Evidence:**

1. `LlavaGuard provides consistent assessments across these examples and continues to demonstrate strong policy-following capabilities, providing well-grounded reasoning using the risk guidelines of the relevant safety category.`
This claim is supported by Figure 2 and 7.

2. `LlavaGuard is the only model that ... as evidenced by its recall performance along with high accuracy.`
This claim is not that rigorous since the experiments are conducted on the held-out test set.
Typically, a model trained on the training dataset might perform better on the held-out test set.
However, since the performance gap between LlavaGuard and baseline methods are large, the claim is still acceptable.

3. `LlavaGuard handles even edge cases effectively.`
This claim is supported by Table 4.

**Essential References Not Discussed:**

LlavaGuard provides assessments that include violated categories and detailed rationales. Therefore, it should add related work about (visual) cot reasoning. For example, one should at least cite `Chain-of-thought prompting elicits reasoning in large language models`.

**Experimental Designs Or Analyses:**

1. The authors compare LlavaGuard with Llava-OV and GPT-4o. However, VLMs like Qwen are also strong and should be compared. Also, can the authors provide performance of visual reasoniong models like OpenAI o1 and QvQ for reference? (It's totally OK if LlavaGuard cannot outperform reasoning models, and it would be amazing if LlavaGuard can.)
2. The authors only use one test dataset for the main experiment. It would be better if there are more.

**Methods And Evaluation Criteria:**

1. In Table 2, the authors use accuracy, recall, and precision to evaluate the model performance. These are very common and widely-used ways for evalustion.
2. In Table 2, the authors also use a metric called PES. The definition of PES is reasonable, but there are two questions: (1) Why not showing PER results in Table 2? (2) Why using the **harmonic** mean of PER and balanced accuracy?
3. In Table 3, the authors conduct a user study. In appendix J, they describe the overall pipeline, which seems to be reasonable.

**Other Comments Or Suggestions:**

1. Line 63, `detailed` is a typo and should be changed to `details`.
2. In Figure 10, the authors don't need to mention the files are `all_data.json` and `test.json`. The `v24` tag is also confusing.

**Other Strengths And Weaknesses:**

1. The idea of providing a rating, category, and rationale is novel in VLM safety. It is relatively well studied in general LLM/VLM.
2. Demonstrating applied use cases is useful.

**Questions For Authors:**

1. Baselines: VLMs like Qwen are also strong and should be compared. Also, can the authors provide performance of visual reasoniong models like OpenAI o1 and QvQ for reference? (It's totally OK if LlavaGuard cannot outperform reasoning models, and it would be amazing if LlavaGuard can.)
2. Benchmarks: The experiments are conducted on the held-out test set.
Typically, a model trained on the training dataset might perform better on the held-out test set.
 It would be better if there are more benchmarks.
3. The authors should add related work about (visual) cot reasoning since adding rationale has been well studied in general VLMs.
4. The authors should fix minor typos in `Other Comments Or Suggestions` section.

I'm happy to raise my score if these questions are well resolved.

**Relation To Broader Scientific Literature:**

1. The paper studies how to build a vision safeguard model, which is important.
2. LlavaGuard goes beyond rigid classifications and provides assessments that include violated categories and detailed rationales. CoT and reasoning have been proved to be effective, and the application of these methods to VLM safety is good.

**Theoretical Claims:**

The paper has no theoretical claims.

---

> ### Author Rebuttal · Authors · 2025-04-01
>
> Thanks for your detailed response and the constructive feedback! Below we address your concerns.
>
> ---
> ### **1. Generalization Beyond the Held-out Test Set**
> While achieving strong performance on the held-out test set is not a small step, we agree that exploring additional datasets provides valuable insights. For this reason, we evaluated LlavaGuard on several real-world (e.g., ImageNet) and synthetic datasets (e.g., Stylebreeder), with users verifying LlavaGuard assessments (see overall score Tab. 3). Below, we break down those scores across datasets. These user evaluations demonstrate that our approach generalizes well beyond the original training and test sets. While individual dataset scores should be interpreted with caution due to low support (e.g., some have only 40 examples annotated), the overall generalization of LlavaGuard remains well-supported.
> ||Rating|Category|Rationale
> |-|-|-|-
> |Stylebreeder|90.5|85.7|76.2
> |COCO|80|80|68
> |ImageNet|84.6|84.6|80.8
> |CC12M|100|100|100
> |GenAI|89.5|89.5|89.5
>
> We have not identified any suitable benchmarks with safety annotations, as most existing ones primarily focus on (conversational) model responses rather than analyzing image content directly (e.g., [MSTS](https://arxiv.org/abs/2501.10057)).
>
> ### **2. Clarifications on PER/PES Metric**
> As explained in Sections 4 (lines 182 ff) and 5 (lines 246-252), we use a combination of PER and balanced accuracy to ensure metric robustness to data imbalance. In a dataset with a dominant class, e.g. "unsafe" samples, there will be more policy exceptions labeled as "safe". A safety classifier that always predicts "safe" will naturally classify many policy exceptions correctly, inflating its PER. However, this can be misleading, as it may give the illusion of greater flexibility than the model actually possesses. To account for this imbalance, we introduce PES, which combines PER with balanced accuracy. PES can be seen as a more reliable measure for policy modifications. That said, we recognize the standalone value of PER (see table below, Response3) and included it in the paper.
>
> ### **3. Comparison with other VLMs**
> Recently, there has been a surge in strong VLMs, and we have now evaluated several models: Qwen2.5, InternVL2.5, and the reasoning model QvQ. All models initially showed lower performance, with Qwen models scoring 67-71%. Moreover, we developed QwenGuard using our framework, once again outperforming their baseline models by a significant margin (bal. acc. of 88-90% and PES of 85%, see Table below), on par with LlavaGuard.
>
> We also investigated building SiglibGuard (based on Siglip2-large) as a lightweight alternative to LlavaGuard. Notably, the Siglib model does not natively support safety classification, meaning there is no direct baseline. Instead, to adapt it for safety evaluation, we train a classification head on top. While SiglibGuard outperforms all VLM baselines, it still lags behind LlavaGuard and other VLM-Guards by more than 16%. A further central limitation is that CLIP-like models cannot inherently process (varying) safety policies. The safety policy provided as input to the VLMs offers a strong and beneficial inductive bias, which CLIP-like methods are missing.
>
> When examining reasoning models, QvQ reaches only 60% balanced accuracy. In analyzing QvQ's failure cases, we observed that it often got lost in recursive reasoning traces, highlighting the complexities of safety evaluation--a promising area for future research. For example, our generated rationales could serve as training data to develop safeguards with safety reasoning traces.
> |Model|Acc|Rec|Prec|PER|PES
> |-|-|-|-|-|-
> |InternVL2.5-1B|50.6|88.1|44.0|6.7|11.8
> |InternVL2.5-8B|61.3|31.3|73.7|70.0|65.3
> |InternVL2.5-78B|66.9|46.1|74.4|66.7|66.8
> |Qwen2.5-VL-3B|68.1|79.7|58.7|20.0|30.9
> |Qwen2.5-VL-7B|67.6|49.2|73.1|60.0|63.6
> |Qwen2.5-VL-72B|70.8|60.0|71.8|52.2|60.1
> |QVQ-72B-Preview|62.0|25.5|**93.4**|**94.2**|74.8
> |GPT-4o|72.9|56.0|81.1|82.2|77.3
> **Fine-tuned Models**
> |SiglipGuard (ours)|73.7|75.6|67.5|24.4|36.7
> |QwenGuard-3B (ours)|88.7|87.8|86.8|81.1|84.7
> |QwenGuard-7B (ours)|89.7|88.9|87.9|80.0|84.6
> |LlavaGuard-0.5B (ours)|88.7|86.7|87.9|85.6|87.1
> |LlavaGuard-7B (ours)|**90.8**|**91.4**|88.0|82.2|**89.9**
>
> ### **4. Related Works on CoT and Visual Reasoning**
> We agree, and this citation ("Chain-of-Thought Prompting Elicits [...]") was actually already included in our bibliography, but accidentally missed including it in the main text. We will revise our related work section about visual CoT and reasoning accordingly and add further relevant references, such as "Measuring and Improving Chain-of-Thought Reasoning in Vision-Language Models", focusing on consistency in visual reasoning.
>
> ### **5. Typographical Errors and Figure Clarifications**
> Thanks, fixed it!
>
> ---
> Thanks for considering our responses. Please also see our responses for the other reviewers. Given our responses, we would appreciate it if the reviewer reconsiders their score.

---

> > ### Comment · Reviewer_LJnY · 2025-04-01
> >
> > I appreciate the additional experiments. I will adjust my score accordingly.

---

> > > ### Author Response · Authors · 2025-04-02
> > >
> > > We want to thank you for the valuable feedback and your prompt response. We are pleased to hear that our additional experiments are well received and agree that they have improved our paper.

---

### Official Review · Reviewer_7XHP · 2025-02-28

**Overall Recommendation:** 4

**Summary:**

The paper introduces a vision-language-based framework specifically designed for safety compliance verification in visual content. It first establishes a context-aware assessment covering nine safety taxonomies and uses it to curate a human-labeled dataset. This dataset includes ground truth safety ratings, violated categories, and human rationales, making it well-suited for content moderation scenarios. The authors fine-tune the Llava model and demonstrate its performance, showing that it outperforms other state-of-the-art VLMs in specific safety tasks. Additionally, they show that LlavaGuard can detect unsafe content in existing natural image datasets such as ImageNet and AI-generated image datasets such as I2P.

**Claims And Evidence:**

Claims are supported by clear evidence.

**Essential References Not Discussed:**

See the strength and weakness session.

**Experimental Designs Or Analyses:**

See the strength and weakness session.

**Methods And Evaluation Criteria:**

The proposed methods and evaluation criteria make sense for the problem.

**Other Comments Or Suggestions:**

No.

**Other Strengths And Weaknesses:**

Strengths:
1. This is the first work leveraging vision-language models (VLMs) for safety compliance verification, marking a significant step forward in the field.
2. The paper introduces a well-structured safety taxonomy covering a broad range of commonly encountered safety rules, with clear definitions provided for each category.
3. A major contribution is the release of a curated dataset with human-labeled safety ratings, violated categories, and rationales, which will be valuable for future research in this domain.
4. The authors fine-tuned an open-source VLM on this dataset, demonstrating its effectiveness in safety compliance tasks compared to native VLMs. Additionally, the release of this fine-tuned model further enhances its impact on the research community.
5. The study includes a thorough analysis of model predictions and extends beyond the curated dataset by evaluating its ability to detect unsafe content in publicly available datasets such as ImageNet and I2P, further demonstrating its applicability and contribution to the field.

Weaknesses:
1. The study focuses solely on fine-tuning Llava, without comparisons to other prominent VLMs such as Qwen or MiniGPT. A broader evaluation, including fine-tuned VLMs and native image-based deep learning models (e.g., ResNet), would strengthen the findings.
2. The proposed taxonomy is fixed and lacks support for dynamic adaptation. While data augmentation is applied to adjust category relevance during training, LlavaGuard does not inherently allow for flexible taxonomy updates. In real-world applications, compliance rules evolve over time, requiring costly and time-consuming retraining for customization.
3. The taxonomy is relatively concise, whereas real-world compliance rules can be much more complex and lengthy. It would be beneficial to evaluate how LlavaGuard performs when handling long-form compliance regulations.
4. Some rule definitions in the taxonomy are vague and subjective. For instance, under O3: Sexual Content, the guideline states that content should not "contain sexually explicit (i.e., erotic) content," which is inherently subjective, even for human annotators. The study lacks an in-depth analysis of how the model handles such subjective definitions and potential biases that may affect safety assessments.

**Questions For Authors:**

1. How do you address the challenge of adapting LlavaGuard to customized real-world safety compliance rules, as highlighted in the weaknesses? Given that compliance regulations evolve over time and vary across domains, what strategies do you propose to enable dynamic taxonomy updates without requiring extensive retraining?

2. What is your vision for extending this work to video compliance scenarios?

**Relation To Broader Scientific Literature:**

The first VLM-based work on safety compliance.

**Theoretical Claims:**

No theoretical claims.

---

> ### Author Rebuttal · Authors · 2025-04-01
>
> Thanks for your detailed response and the constructive feedback! Below we address your concerns.
>
> ---
> ### **W1 Additional VLMs**
> According to your suggestion, we included additional prominent VLMs (e.g., Qwen-VL) and baseline image-centric models (SigLip2), please refer to Response3 for LJnY.
>
> ### **W2,W3,Q1 Future strategies to adapt LlavaGuard to real-world safety rules**
> While LlavaGuard already exhibits preliminary adaptability (e.g., through our policy-driven inference), we acknowledge that customizing it to evolving, real-world compliance regulations across diverse domains presents important future challenges. To effectively enable more complex policy changes without requiring resource-intensive retraining, we envision the following strategies:
>
> 1. **Rule-Based Classification.** Modular, rule-based classifiers that operate on top of the VLM’s intermediate outputs--such as structured rationales or even extracted semantic symbols from rationales. These symbols can serve as inputs to a logic-based reasoning layer (e.g., a logic program), which can be updated based on evolving safety taxonomies.
> 2. **Inference-Time Reasoning.** Novel scaling methodologies beyond model parameters can offer significant benefits for adaptability. For instance, scaling inference-time compute to incorporate step-by-step reasoning traces can enhance nuanced safety understanding and, in turn, allow more nuanced adaptations of safety specifications.
>
> ### **W4 Subjective Rule Definitions.**
> This is indeed a crucial concern. Subjectivity of (safety) taxonomies is a well-known challenge, not only in recent AI safety taxonomies but also in other real-world applications and guidelines, such as [PEGI](https://pegi.info), or similar frameworks, where classification rules require subjective interpretation. For example, PEGI’s treatment of simulated gambling recently led to a controversial age rating that was later successfully appealed (see [IGN article](https://ign.com/articles/balatro-dev-successfully-appeals-pegi-18-rating-over-simulated-gambling)), highlighting challenges of subjectivity in the realization of such taxonomies.
>
> A key motivation for leveraging VLMs like LlavaGuard is their capability to handle both precisely defined and inherently subjective or vague safety guidelines through their acquired commonsense understanding. However, we recognize that subjectivity remains a challenge—LlavaGuard likely reflects some averaged subjectivity embedded in its training data rather than an objective standard.
>
> Though we did not directly analyze LlavaGuard’s handling of subjective guidelines, results from our user study demonstrate strong human-model agreement—including subjective and ambiguous cases—indicating that LlavaGuard’s assessments generally align with human interpretations.
>
> However, we acknowledge that biases embedded within the model’s acquired commonsense understanding might influence specific subjective decisions. We agree that systematically investigating, understanding, and mitigating these biases is an essential direction for future research. We will add this to our discussion.
>
> ### **Q2 Extension to Video**
> Thank you for highlighting this important direction. Technically, LlavaGuard could be adapted to video scenarios using a sliding-window approach, processing videos frame-by-frame or through short segments, and aggregating frame-level safety assessments into an overall video compliance score.
>
> However, videos inherently combine multiple modalities, most notably audio alongside visual content. Therefore, a more robust approach would involve extending our current vision-language models (VLMs) to multimodal architectures capable of jointly modeling visual, textual, and auditory signals.
>
> A particularly relevant and impactful application could involve compliance verification for video games, films, and streaming content, which currently rely on established rating frameworks such as PEGI or ESRB. Our vision includes developing multimodal safeguards capable of automating and augmenting traditional human-based rating processes, ensuring consistent and transparent safety assessments at scale. Additionally, with recent advancements in generative AI, we anticipate the emergence of interactive applications where generative models dynamically create video clips in real time, based on user prompts or interactions. In such applications, real-time safety checks become crucial, as the dynamic nature of generated content could rapidly introduce risks that traditional compliance approaches might not detect. LlavaGuard, due to its robust multimodal understanding and flexible policy adherence, and follow-up advancements, could play an essential role in monitoring, assessing, and ensuring the safety of these generative interactive media environments.
>
> ---
> Thanks for considering our responses. Please also see our responses for the other reviewers. Given our responses, we would appreciate it if the reviewer reconsiders their score.

---

> > ### Comment · Reviewer_7XHP · 2025-04-01
> >
> > Thanks for addressing all my comments. I have no further concerns. I would raise my score to Accept. A good paper.

---

> > > ### Author Response · Authors · 2025-04-02
> > >
> > > We would like to thank you for the constructive feedback and the prompt response. We are pleased to hear that we have addressed all your comments and that there are no further concerns. We believe your feedback has further enhanced our paper.

---

### Official Review · Reviewer_4Cus · 2025-03-09

**Overall Recommendation:** 3

**Summary:**

- Key contribution: This paper presents a safety guard suite, LLavaGuard, with a dataset consisting of ~5K images annotated with safety labels and rationales, and two models trained using the dataset.

- Motivation: The key motivation behind this is that safeguard models and datasets are rare in the visual domain despite some previous limited attempts, such as LAION-nsfw classifiers. The authors claim that we need a more comprehensive coverage for better moderating images.

- Dataset Curation: The taxonomy is built according to AI regulations, including nine safety categories. Images are sourced from SMID and supplemented via web-scraped images. Human annotators are employed to label the image according to safety risk taxonomy, and models are prompted to generate rationales explaining why the images are classified. Data augmentation is introduced to cast the classification problem into One vs All (non-violating) formats for better adaption.

- Empirical findings: Two models based on LLaVA-OV-0.5B / 7B are trained using the proposed dataset, where the LLaVAGuard7B model shows better classification accuracy compared with previous moderator models such as OpenAI-omni-mod.
Further analysis applying the LLaVAGuard to ImageNet shows wrongly labeled images in the original dataset. LLaVAGuard can also safeguard generative models such as StableDiffusion, achieving a high agreement > 80% with human users.

**Claims And Evidence:**

The claims in the paper are generally well-supported.

**Essential References Not Discussed:**

N/A

**Ethical Review Concerns:**

- There are no details of the human annotators, which might lead to biased annotation results;
- The web-scraped images might have copyright issues.

**Ethics Expertise Needed:**

["Discrimination / Bias / Fairness Concerns", "Legal Compliance (e.g., GDPR, copyright, terms of use)"]

**Experimental Designs Or Analyses:**

- The experiments for filtering ImageNet and safeguarding generative is generally sound.

- I am curious about the failure cases of the current guard models. any categories in which they might fail more frequently? Also, providing some false-positive samples detected would be helpful?

**Methods And Evaluation Criteria:**

- The dataset curation section lacks essential information about human annotators, including demographics, total number, compensation, and procedures for handling diverse perspectives on subjective safety guidelines. The authors should address how conflicting decisions and ambiguous cases were resolved. Additionally, questions remain about copyright compliance for web-scraped training images and the predominantly Western-centric regulatory framework that fails to account for cultural nuances.

- Table 2 reveals that scaling the guard model from 0.5B to 7B yields minimal performance improvements. This raises the question of whether simpler approaches, such as fine-tuning CLIP/SigLIP 2, might achieve comparable results despite lacking rationale generation capabilities—a comparison that would strengthen the baseline evaluation.

- The methodology for rationale generation requires clarification regarding which models were employed and what quality assurance measures were implemented to ensure accurate and helpful explanations.

- Finally, the evaluation would benefit from including results across a more diverse set of models, particularly examining how LLama-Vision 11B or Qwen-VL series models might perform when trained on the curated dataset.

**Other Comments Or Suggestions:**

N/A

**Other Strengths And Weaknesses:**

Cons: (minor) Despite the efforts of curating the dataset is well-appreciated and the significance of the safety moderation is well-recognized, this paper does not bring any technical contribution to the community.

**Questions For Authors:**

- What is the downstream task performance of the models trained using the filtered dataset? e.g., would the accuracy of an image classifier trained on the filtered ImageNet become lower than that trained on the original dataset?

====

The additional annotationdetails and updated comparison with recent models provided during rebuttal effectively address my concerns, therefore I am increasing my scores from 2 to 3.

**Relation To Broader Scientific Literature:**

The paper extends LLaMAGuard and LAION-NSFW using curated dataset to train a LLaVAGuard for image safety moderation.

**Theoretical Claims:**

N/A

---

> ### Author Rebuttal · Authors · 2025-04-01
>
> Thanks for your detailed response and the constructive feedback! Below we address your concerns.
>
> ---
> ### **1. Annotators and Dataset Information**
> We reaffirm our commitment to ethical and regulatory standards. To ensure annotators' well-being, dataset annotation was directly performed by the authors, and our user studies included researchers with expertise in handling safety-related content, following annotator protection guidelines by [Vidgen et al. (2019)](https://aclanthology.org/W19-3509). For more details, please refer to App. J.
>
> Regarding the dataset, all annotators are male, White, and between 20-40 years old. We adopted a prescriptive annotation approach, collaboratively developing a taxonomy that defines a detailed categorization. For edge cases, all annotators discussed potential contradictions to achieve consensus. For the user study, participants were aged 20-30, 30% male and 70% female, with 70% identifying as White (from the US and Europe) and 30% as Asian-American. Despite the demographic differences between dataset annotators and user study participants, we observed a high level of agreement.
>
> Regarding copyright compliance, we adhere to Fair Use principles. Efforts have been made to ensure that images are in the public domain or under open licensing. Most of our data is based on SMID (Creative Commons license). For the remaining data we collected, neither the images are distributed, nor does the model distribute data. Consequently, training our model on this data does not infringe copyright, as it serves a non-commercial research purpose.
>
> We will incorporate these clarifications to enhance transparency.
>
> ### **2. Classifier Baselines**
> We evaluated simpler approaches like CLIP-based classifiers (Q16, App. Tab. 5), achieving significantly lower performance (only 69% acc.). We have also finetuned SigLIP2-large, which reached 99.6% train and 73.7% test acc. (more details in Response3 to LJnY). While the performance is higher than Q16's, it still lags behind LlavaGuard's accuracy by more than 16%. The main reason is that CLIP-like models cannot handle (varying) policies. Testing SigLIP under fixed-policy conditions yielded improved acc. (79%), still far below LlavaGuard.
>
> ### **3. Rationale Quality**
> We acknowledge that a quality validation of guided rationales strengthens our findings. Therefore, we present a more comprehensive evaluation using GPT-4o to compare guided vs. non-guided rationales across our entire dataset. The GPT-4o judge scored rationales based on comprehensiveness, accuracy, and guideline adherence:
> |Llava-34B|Guided Rationales|Base Rationales
> |-|-|-
> |Mean Quality Score (1–10)|**9.1**|3.8
> |Median Quality Score (1–10)| **9.0**|3.0
> |Win Rate (%)|**99.9**|0.1
>
> The results demonstrate that our guided generation approach produces substantially better rationales, which is supported by qualitative examples in App. Fig. 7. Additionally, we benchmarked rationale quality across Llava model scales:
> |Model|Win Rate (%)|Mean Score (1-10)|Median Score (1-10)
> |-|-|-|-
> |Llava-34B|**83.9**| **9.0**|**8.4**
> |Llava-13B|9.6|7.0|6.7
> |Llava-7B|6.6|7.0|6.8
>
> We used Llava-34B in our final setup to generate rationales for LlavaGuard training and the table shows it largely outperforms the other base models. These clarifications have been added to the paper.
>
> ### **4. Additional VLMs**
> We fully agree about the need for a broader evaluation range, particularly examining strong recent models (e.g., Qwen-VL series). Please see our detailed Response3 to LJnY.
>
> ---
> ### **5. Failure Case Analysis**
> Upon examination of false-positives, we noted common failure cases of LlavaGuard:
>
> - Images near decision boundaries proved inherently challenging.
> - The model occasionally outlined challenges interpreting embedded image text. For instance, it misclassified signage explicitly prohibiting sexual harassment of nurses as unsafe content.
>
> We uploaded an [overview](https://anonymous.4open.science/r/LlavaGuard-FE7F/figs/false_positives/FP_overview.md) (Please note some samples might be disturbing). We add these insights to our paper.
>
> ### **6. Downstream Performance on Filtered Dataset**
> We trained ResNet50 (50 epochs, LR=0.001, AdamW optimizer) from scratch on original vs. LlavaGuard-filtered ImageNet (~20K images removed, ~1% overall).
> |overall|Top-1 Acc.|Top-5 Acc.
> |-|-|-
> |ImageNet|67.2±0.5|87.2±0.6
> |ImageNet-filtered|67.4±0.4|87.1±0.6
>
> Despite removing up to 55% of samples in some classes (e.g., “assault rifle,” “army tank,” “missile,” “syringe”), overall downstream performance remains unchanged. Inspecting the most filtered classes only, we observe a gap.
> |top-10 most filtered classes|Top-1 Acc.|Top-5 Acc.
> |-|-|-
> |ImageNet|53.6±3.5|82.4±3.1
> |ImageNet-filtered|46.9±4.8|77.3±3.6
>
> This shows that careful safety filtering does not have to come with compromises in downstream performance.
>
> ---
> Thanks for considering our responses. Given our response, we would appreciate it if the reviewer reconsiders their score.

---

> > ### Comment · Reviewer_4Cus · 2025-04-02
> >
> > Thank you for your response which effectively addresses my concerns. I will increase my scores accordingly :)

---

> > > ### Author Response · Authors · 2025-04-02
> > >
> > > Thank you for the constructive feedback and the quick response. We are pleased to have effectively addressed your concerns. Your insights have been valuable and have helped us further refine our paper.

---

### Decision · Program_Chairs · 2025-05-01

**Decision:**

Accept (poster)

**Comment:**

All reviewers have provided positive scores for this submission, highlighting its strengths in key contribution and extensive experiments. Given the unanimous positive feedback and the recognition of its contribution to the area, the AC carefully reviewed the paper and concurred with the reviewers' assessments, therefore supporting the decision to accept this submission.